# Tango-Therapy Intervention for Older Adults with Cognitive Impairment Living in Nursing Homes: Effects on Quality of Life, Physical Abilities and Gait

**DOI:** 10.3390/ijerph20043521

**Published:** 2023-02-16

**Authors:** Lucía Bracco, Clara Cornaro, Arrate Pinto-Carral, Sabine C. Koch, France Mourey

**Affiliations:** 1Inserm U1093-Cognition, Action and Sensorimotor Plasticity, Faculty of Sport Sciences, University of Burgundy, 21078 Dijon, France; 2Research Institute for Creative Arts Therapies (RIArT), Alanus University of Arts and Social Science, 53347 Alfter, Germany; 3SALBIS Research Group, Faculty of Health Sciences, Nursing and Physiotherapy Department, Universidad de León, 24401 Ponferrada, Spain; 4Department of Therapy Sciences, SRH University Heidelberg, 69123 Heidelberg, Germany

**Keywords:** older adults, cognitive impairment, tango therapy, quality of life, well-being, physical performance, walking performances, abilities of daily living, dance movement therapy, arts with therapeutic intent

## Abstract

Cognitive impairment in older adults is associated with poor gait performance, physical decline, falls and poor quality of life. This paper analyzes the feasibility and efficacy of tango-based intervention in older people living in nursing homes with and without cognitive impairment. A multicenter study, with pre- and post-test, was carried out. Intervention attendance, well-being, physical abilities (short physical performance battery), walking performance, functional capacities (Katz Index) and quality of life (quality of life in Alzheimer’s disease) were assessed. Fifty-four participants (84.9 ± 6.7 years, mini mental state examination 14.5 ± 7.4) completed the protocol. Intervention attendance was 92%, and the mean subjective well-being after each session was 4.5 ± 0.5 (on a five-point scale). A statistically significant improvement was found in the quality of life (*p* = 0.030). Non-statistically significant changes were found in walking performance (*p* = 0.159), physical abilities (*p* = 0.876) and in functional capacities (*p* = 0.253). This study shows feasibility and suggests evidence for the effects of tango therapy on well-being and quality of life. Further studies are necessary to contrast these findings and to support the role of tango interventions as a holistic approach to prevent functional decline in older people with cognitive impairment.

## 1. Introduction

Cognitive impairment (CI), even at an early stage, is associated with gait disorders, poor physical function and a high risk of falling [1]. It has been shown that CI is a common condition in people living in nursing homes, and this place of residence has been highly linked to physical inactivity, greater risk of sarcopenia, cognitive decline, poor quality of life (QoL) and depression [2].

Evidence suggests that the impoverished environment of long-term care institutions could be an accelerating factor of decline. This is why, to prevent deterioration, it is necessary to provide enriched interventions, involving multisensorial activities, attentional tasks, strength and dynamic balance exercises that promote immediate memories, working memories and praxis [3]. Music and dance are used as non-pharmacological therapy and have been shown to be effective for different conditions, such as mental illnesses, Parkinson’s disease [4] and other neurological conditions [5]. It could be a helpful intervention, particularly if verbal communication is difficult or impossible. In addition to stimulating physical abilities, music and dance are a socio-cultural experience. People come together, create a specific community atmosphere and share an aesthetic space [6]. Dance may improve QoL by simultaneously engaging the sensory systems and by stimulating physical, emotional and cognitive functions [7].

Tango is assumed to be particularly helpful for improving mobility, motor–cognitive function and gait in independent older adults [8]. This activity of moderate intensity that implicitly invites exercise through rhythm, imitation and synchronization could be positioned as a promising intervention in frail older people with CI. Its regular practice can have a positive impact on cognitive abilities, as it requires sustained attention, interactional sensitivity, sensorial presence, spatiotemporal awareness and use of memory. Studies on tango for Parkinson’s disease have shown its effectiveness, particularly in ameliorating QoL, as well as improving psychomotor and cognitive functions [9]. Hackney and Earhart showed that tango strengthens QoL under different conditions [10,11]. They found that partnered (vs. non-partnered) tango and long-term (vs short-term) tango, such as a year group, are particularly effective [12]. The year group in the study of Duncan and Earhart strengthened not only QoL, but also the reuptake of hobbies, such as gardening and restaurant visits with a partner [13]. Tango offers an enriched environment; through use of the different senses, i.e., touch, hearing and vision, the participant experiences being part of a group. After a while, music and dancing can activate memories and create a sense of safety and pleasure [14].

From a biomechanical and kinesthetic perspective, tango is a dance based on normal gait [15] and its particular characteristics could be of great help in the gait rehabilitation of older adults with CI. Single-limb support is the basic position, which requires constant work on static postural control. Dynamic postural control is stimulated by the intrinsic imbalance generated by all movements performed with the free limb. The anticipation of body weight transfer is accentuated due to the need to give a signal to the partner to generate a synchronous movement. At each step, the malleolus rubs against each other, reducing the distance between the feet, which is usually increased in older people with balance impairment. Changes in the rhythm of music induce a change of gait speed, forcing the dancer to move in a varied spectrum of velocities and to continuously modify step length. From these characteristics, tango appears to be an excellent tool for the rehabilitation of balance and gait [16], as well as for the prevention of functional decline [17].

There is limited evidence of the feasibility and efficacy of dance intervention for populations of older people with CI. Few of the previous studies include institutionalized persons of advanced age and with dementia [18]. Trials of higher methodological quality, large sample sizes and clarity of intervention are needed to evaluate dance movement therapy in this population [19]. In a literature search, we did not find any survey that evaluates tango in a similar population. This is the first study to analyze the feasibility and effectiveness of tango therapy on QoL, physical abilities and gait in older people living in nursing homes, with and without CI.

## 2. Materials and Methods

This multicenter study, with pre- and post-intervention assessment, was carried out in eight nursing homes between October 2021 and March 2022 in France during the COVID-19 crisis. The absence of a control group was due to the lack of funding to implement a control intervention and due to the ethical issues that are inherent to a passive control group.

### 2.1. Participants

The participants were all over 65 years old, independent in ambulation with or without a walking aid. They agreed to participate and lived permanently in one of the nursing homes where therapeutic tango interventions were implemented. The exclusion criteria were medical contraindication, limited life expectancy and bedridden persons. The study exit criteria were withdrawal from the participation agreement, participation in less than 50% of the tango sessions, psychiatric state, pathologies and care no longer allowing for the continuation of the study, and death.

### 2.2. Ethics

The study was explained to the participants, caregivers and families, and information forms were distributed. Participants were enrolled in the study after consent was obtained. The study was conducted in accordance with the Declaration of Helsinki and approved by the Research Ethics Committee of the University of Burgundy (CERUBFC 2021-09-15-026).

### 2.3. Intervention

The intervention was implemented by ABB Reportages (To know more: http://www.abbreportages.fr/content/view/214/186/lang,english/ (accessed on 1 December 2022)), an independent film production studio working on the origins, sense and the power of music. Participants attended a 1 h tango session, once a week for twelve weeks, carried out by nursing staff who previously received training in therapeutic tango at the University of Burgundy (To know more: https://sefca.u-bourgogne.fr/toute-lactualite/379-la-melodie-d-alzheimer.html?fbclid=IwAR0bu8hkfWfMJ39oRRZNGcI9aFG-HK4CTy9YucKr72Quwvb04P4ivC4TkEE (accessed on 1 December 2022)). The training was based on tango, which is a completely improvisational form of partnered dance. The educators were trained in the Sistema Dinzel, which is an approach and teaching of tango, particularly suited to therapeutic intent. For founders Gloria and Rodolfo Dinzel, improvisation and connection are at the heart of tango. Hence, the aim is to provide enough structure and tools to improvise and connect, empowering dancers to create their own experience. This approach lends itself to therapeutic implementation as it is free of a strict form of ‘right’ and ‘wrong’ and rather focuses on the creative experience and being in connection with the music and the partner [20].

A dance movement therapist (DMT) and a musician accompanied the interventions twice a month, while the rest of the sessions were staff-led (with a “train the trainer/carer” idea). The aim of the attendance of the DMTs and musicians was to guarantee a professional quality. By being there, connection to the project was sustained, while still fostering autonomy for the nursing staff to facilitate their own bi-weekly interventions with the new input they received. This structure also provided an opportunity for the nursing staff to ask questions if there were any challenges or uncertainties in carrying out the intervention. An overview of the tango therapeutic session content is shown in Table 1. Within each session, facilitators were encouraged to hold a safe space, stimulate social connection and teach new tango skills while meeting the individuals where they were. The main part of the intervention was left open and was facilitated depending on the state of the participants. While the choice of tango music varied slightly depending on the facilitator, some songs were frequently used for group singing by all groups (Table 2). Depending on the exercise, the musician played milonga, waltz or tango to assist in the task.

### 2.4. Outcome Measures

One member of staff at each center was instructed to perform the assessments in collaboration with the main investigator. Sociodemographic data and medical history of each resident were collected. Cognitive performance was assessed through the mini mental state examination (MMSE) [21]. A score higher than 25 indicated an absence of CI, between 21 and 25 indicated mild CI, between 11 and 20 indicated moderate CI and lower than 11 indicated severe CI [22]. During the statistical analyses, the categories were simplified into severe–moderate CI (SMCI) (MMSE 0–20) and mild or no CI (MNCI) (MMSE 21–30). Overall comorbidity was assessed using the Charlson Index of Comorbidity (CCI) [23]. The participants were assessed before and upon completion of the twelve-week intervention. The subjective feeling of well-being was evaluated after each session. Intervention attendance was recorded and was calculated as follows: ([number of dance sessions attended/total number of dance sessions] × 100).

The subjective feeling of instantaneous well-being was assessed after each session through a visual analog scale of well-being named EVIBE (Échelle d’évaluation instantanée de bien-être, Scale of instantaneous well-being). On a graduated ruler from 1 to 5, the participants positioned their feeling of well-being in response to the question “How do you feel now?”. An answer of “1” corresponded to the weakest feeling of well-being and an answer of “5” corresponded to the strongest feeling of well-being. This scale was previously validated in older people with severe dementia, showing a high level of intrajudge reliability (intraclass correlation between 0.79 and 0.90; *p* < 0.001) [24].

Physical performances were measured by the short physical performance battery (SPPB) with three evaluation criteria as: balance; walking speed; and sit to stand [25]. Specifically, during the balance test, the subject had to maintain each of three distinct positions for 10 s (feet together, semi-tandem and tandem). Failure of a step was the condition to start the second test, in which the subject walked 4 m two consecutive times. To perform it, the participant walks at their normal pace and may use an assistive device if needed. The participant walks down a hallway through a 1 m zone for acceleration, a central 4 m “testing” zone and a 1-meter zone for deceleration (the participant should not start to slow down before the 4 m mark). The shorter time was retained. The score varied according to the gait speed as follows: 4 (>0.83 m/s); 3 (0.83–0.64 m/s); 2 (0.64–0.46 m/s); 1 (<0.46 m/s); or 0 (not attempted). Finally, the time taken to complete the fastest five chair lifts without the help of the upper limbs was evaluated. At the end of the three tests, a score of a maximum of 12 points could be obtained [26]. In addition to gait speed, walking performances were assessed based on the participants’ need of walking aids (without aids, including canes, crutches or rollators).

Functional capacities were assessed using the Katz Index [27], consisting of a questionnaire assessing abilities in six activities of daily living (ADL) as follows: personal hygiene care; dressing; toilet use; locomotion; continence; and eating. For each domain, the answer varies between 1 (complete independence), 0.5 (partial independence) or 0 (absolute dependence). In total, an index of zero to six is obtained, where zero indicates the highest degree of dependence.

QoL was assessed using the QoL-AD French version [28]. This questionnaire was administered directly to the participant, up to a severe stage of the disease, and to the main caregiver. The participant and caregiver ratings were combined into a weighted composite score as follows: (2 × patient score + 1 × caregiver score)/3. Thus, the answers provided by the patient remained preponderant in this model. The QoL-AD comprises 13 items (physical health, energy, mood, living situation, memory, family, marriage, friends, self, ability to carry out daily tasks, ability to do things for fun, money and life as a whole). Response options included 1 (poor), 2 (fair), 3 (good) and 4 (excellent), for a total score of 13–52, with higher scores indicating a better QoL. This questionnaire was validated in French showing good internal consistency (Cronbach’s alpha coefficient > or =0.70) and good reliability (intraclass correlation > 0.80) at a 2-week interval for the patient and caregiver questionnaires [29]. Cronbach’s alpha coefficient for test–retest reliability in older people with severe CI was 0.8930. There was a significant correlation between the QOL-AD and the Activities of Daily Living Inventory (ACDS-ADL) (*p* < 0.001) and with the Health Status Questionnaire (HSQ-role-physical) (*p* < 0.01). The QoL-AD is valid and reliable in people with an MMSE score of >2 [30].

### 2.5. Statistical Analysis

The relative frequencies of the qualitative variables were determined. For the quantitative variables, the means and standard deviations (SDs) were calculated with 95% confidence intervals (95% CIs). The Shapiro–Wilk test was used to determine data distribution normality. Quantitative outcome measures taken before and after intervention were compared using a paired t-test for parametric data distribution and Wilcoxon signed-ranked test for non-parametric data distribution. A Mann–Whitney U test was used to compare quantitative outcome measures between the different categories of CI. A Chi-squared test was used to compare the distribution of categorical variables before and after the intervention. The alpha level was set to *p* < 0.05. Statistical analyses were carried using SPSS version 25 from IBM.

## 3. Results

### 3.1. Sample Characteristics

As shown in Figure 1, out of 102 people screened for eligibility, 74 were recruited from 8 different nursing homes for a 12-week tango intervention, according to predefined inclusion criteria. Fifty-four participants completed the intervention program and were therefore eligible for statistical analyses. Reasons for dropout were change of care structure (n = 1), a deterioration in their state of health (n = 4), death (n = 1) and absence in more than 50% of the sessions (n = 14). No significant differences in baseline characteristics were found between dropouts and those participants who completed the intervention. The mean age was 84.9 ± 6.7 years old, and the average MMSE score among 54 participants was 14.5 ± 7.4, with 52% showing moderate CI and 24% showing severe CI. Table 3 shows the main characteristics of the total sample by category according to the MMSE score.

### 3.2. Feasibility

During this study, 67 sessions were carried out among the 8 nursing homes, with an average per center of 8.7 ± 1.6 sessions. Each participant completed an average of 8 ± 1.7 sessions. Average attendance was 92% and the subjective feeling of well-being after sessions measured with EVIBE was 4.5 ± 0.5 (on a five-point scale).

### 3.3. Outcomes

The main results are presented in Table 4. A statistically significant improvement in QoL was noted (*p* = 0.030). No statistically significant changes were found in the participants’ physical performances or their ability to perform ADLs. There was a small improvement in SPPB, especially in the balance subscore, but there was also a small decrease in functional abilities. There were also no statistically significant changes in walking performance (*p* = 0.159).

#### 3.3.1. Intergroup Comparison

Table 5 shows outcomes by categories according to the participant’s MMSE score. At the baseline, the groups presented statistically significant differences in the sit-to-stand test (*p* = 0.042) and in the Katz Index (*p* = 0.003). At post-test, this difference was more evident and intensified due to the divergence between the two groups in gait speed (*p* = 0.020) and in total SPPB score (*p* = 0.025). However, no significant differences were observed in terms of intervention attendance (*p* = 0.084) and in well-being after each session (*p* = 0.087).

#### 3.3.2. Intragroup Comparison

The results show a stabilization of all physical and functional variables despite the social isolation measures imposed by the COVID-19 pandemic. Both groups improved balance, and the MNCI group improved gait speed and total SPPB score. However, there was a slight deterioration in the ability to perform ADLs, but this was not statistically significant and was more noticeable in the SMCI group. Finally, QoL improved in both groups, with a statistically significant improvement in the global score of the SMCI group (*p* = 0.038).

## 4. Discussion

The objective of this study was to analyze the feasibility and effectiveness of tango-based interventions for older people living in nursing homes with and without CI. The results demonstrate that the intervention was both feasible and highly appreciated among the participants. The average attendance rate of 92% and the high rating of subjective well-being after each intervention (4.5 ± 0.5 over 5) show high acceptance. Moreover, tango interventions were effective in improving QoL and maintaining physical and functional capacities. These results are remarkable, especially considering the deterioration in older adults caused by lockdown and other restrictions imposed to contain the spread of COVID-19 [31].

Regarding the characteristics of the population, it is important to highlight the differences between the categories of cognitive performance. Even though the age and gender distribution were the same in both groups, the SMCI group presented greater comorbidities, precarious balance, slower gait speed and poorer abilities to carry out activities of daily living. This well-known association between cognitive performance and motoric function in older people [32] implies that the results of the interventions could be less evident in people with greater CI, but not less valuable because of that.

According to the findings of this study, tango interventions are a beneficial strategy to improve QoL and to reduce the difference between caregivers’ and participants’ QoL perception. Consistent with prior surveys, we found that participants provide better scores on their baseline QoL than caregivers [33,34]. One reason may be social representations of old age and a health culture that is more focused on health-related aspects than on overall QoL [35]. The improvement in caregivers’ assessment after the intervention may be explained by a change in their perception through the relationship created during the tango sessions. Engaging staff in dance interventions might result in improving job satisfaction, and in any case provide staff with the opportunity to engage with residents in an enjoyable way [36]. Previous studies report a positive change in group dynamics after a dance-therapy-based intervention among residents and staff [37]. Tango could contribute to a positive work environment, directly impacting the caregiver’s perception of the patient.

Further we found a significant improvement in the QoL-AD global score, which supports the answers obtained from the participant and the caregiver. The analysis by categories shows a statistically significant improvement in the SMCI group. These findings agree with previous studies, which showed a beneficial effect of dance interventions on the QoL of persons with CI, independently of benefits in physical abilities [38,39]. However, to date, there is a gap in the literature concerning the mechanisms by which music and dance can contribute to improvements in the QoL of older persons with dementia. Dance integrates several brain functions, such as those connected with kinesthesia, musicality and emotion, inducing expressive brain plasticity and structural changes [40]. The sensory stimulation produced through dance seems to be the origin of these changes [41].

Interventions based on tango focus on the work of emotions linked to interpersonal relationships. One of the approaches presented by Cohen-Mansfield to frame non-pharmacological interventions aimed at people with dementia is based precisely on relief of unmet needs, caused by sensory deprivation, boredom or loneliness [42]. Tango interventions are rich in sensory stimuli, provide fun and feelings of connection, probably representing the keys to improving QoL. Furthermore, in a more specific way, tango allows, through the movement of the body and the embrace (“abrazo”), a better awareness of one’s own axis and that of another, thus promoting bodily, relational and emotional discoveries [14]. The maintenance of links and relationships with caregivers, as well as access to meaningful activities, have been identified as factors that exert an influence on the QoL of institutionalized older people [43]. The combination of sensory enrichment, physical activity and social interaction during dance, could explain the positive effect on QoL found in this study. However, further studies with a control group and analyses of different domains of QoL are necessary to understand the mechanisms of this relationship.

Regarding physical and functional abilities, changes were non-significant. Tango interventions could have contributed to maintaining physical capacities during this period of harsh restrictions, representing a positive result. This study took place during the fifth epidemic wave of COVID-19. The measures applied to limit the spread of the virus had direct deleterious effects on the geriatric population, such as sarcopenia and depression, as well as indirect effects through the delay, or even the absence, of care for disease. The impact was even greater among nursing home residents, showing a significant functional, cognitive and nutritional decline in COVID-19 and non-COVID-19 patients [44].

Conversely, the participants in our study did not experience significant deterioration. Other studies with a control group that evaluated the effect of a dance intervention in a similar population have shown a deterioration in the control group and a slowed down deterioration of functional status in the intervention group [45,46]. These findings could provide evidence of a stabilization of physical and functional performance produced by dance interventions.

Concerning gait, there were not significant differences between the groups at the baseline. Although there were no significant changes in gait speed after the intervention, the intergroup post-test analyses showed a significant difference in favor of the MNCI group. Studies have already demonstrated that older people with dementia have altered gait parameters, such as reduced gait speed [1]. This parameter has been proposed as an essential marker in geriatric assessment and is a strong predictor of adverse outcomes, including hospitalizations, disability, cognitive decline and death. Even though the observed changes were not statistically significant (a small improvement in the MNCI group and a small decline in the SMCI group), they may be clinically significant, because it has been suggested that a variation of 0.1 m/s in gait speed may increase mortality by 20% [47].

A large part of the tango interventions focused on walking. Several studies on Parkinson’s disease have observed a significant improvement in gait immediately after a tango lesson [48] after a short-term adapted tango program [49] and after a long-term tango intervention [13]. A study examined the functional changes in the activity of the cerebral areas involved in a locomotor imagery task after one week of training consisting of performing basic tango steps. The results showed an expansion of active bilateral motor areas and a reduction in visuospatial activation in the posterior right brain, suggesting a decreased role of visual imagery processes after intervention in favor of motor-kinesthetic ones [50]. These findings translated into clinical practice could result in significant gait improvement in older people and a reduction in fall risk.

In this way, tango could be an innovative approach to gait rehabilitation because its practice involves multiple exercises of classic rehabilitation. However, instead of being performed analytically, they are performed simultaneously, thus achieving the benefits of functional physical therapy [51], while presumably having a greater motivational effect. However, to obtain real results in terms of tango as a rehabilitation method, it would be necessary to apply the intervention individually. It thus would be possible to personalize the session, emphasizing the specific aspects that must be dealt with. Further studies about the effect of tango in spatiotemporal gait parameters are necessary to understand the mechanisms and to improve this promising method. 

Therapeutic tango is an intervention inspired by the principles of dance movement therapy and rehabilitation for frail older people with CI. Its objective is to provide alternative and complementary therapy to people living in nursing homes. From on-site spontaneous testimonies, it emerged that participants, caregivers and families intuitively perceived an impact on the well-being and on the neuromotor capacities. Staff–resident dynamics improved and relationships between residents were fostered. Walking, which tends to be slowed down with small steps, is transformed by the effect of the music. Despite this being an empirical finding, based on the results and on these spontaneous expressions, we postulate that therapeutic tango could be an effective strategy to prevent functional decline and to improve QoL in older people with CI.

The present study has several strengths. This is the first study to analyze the effectiveness of therapeutic tango in older people with CI living in nursing homes. There are few studies on non-pharmacological therapies in similar populations [18,19] because research in this context presents several challenges and methodological issues. Therefore, one of the strengths of this project is that it yielded an assessment of older participants regarding the effects of an intervention designed specifically for them, thus contributing to the evidence base for the care of older adults [52].

This study also has several limitations. First, the biggest limitation is the lack of a control group. Tango could play a role, no longer improving, but maintaining physical and functional capacities. However, without a randomized controlled trial (RCT), this remains a hypothesis. Additionally, taking into consideration the recommendation of the World Health Organization on exercise for older people with reduced mobility [53], the frequency of tango interventions would be insufficient to produce significant changes in physical abilities. Finally, even if most studies used the SPPB to assess physical performances in older adults with CI, this may not be sensitive enough to measure outcomes because minimal cognitive performance is required to understand instructions and respond appropriately.

The lessons from these limitations have been incorporated into further projects of the research team, namely an RCT conducted to compare tango interventions with other activities that have already been tested with a higher number of interventions (2 or 3 per week), the technical implementation of assessment of spatiotemporal gait parameters, and an evaluation of the risk of falls. Additionally, the mechanisms of QoL improvement must be investigated to be able to gear intervention in an optimal way to the patient.

## 5. Conclusions

Tango therapy has shown to be feasible and is highly accepted to support well-being in older adults living in nursing homes. The results of this study suggest that tango-based interventions have a positive impact on participants’ QoL, and especially in the caregivers’ view of the participants’ QoL. Concerning physical abilities and walking performances, tango therapy could play a stabilizer role, but new studies are needed to establish this relationship. Tango therapy may involve older persons and be particularly beneficial for those with CI. The results obtained here encourage us to continue the search towards understanding the mechanisms that are set in motion through tango and its relationship with the QoL and physical capacities of older adults with CI. Tango therapy with its versatile sensory–motor possibilities may be just the intervention needed for older adults in nursing homes to prevent decline and to foster joy and well-being.

## Figures and Tables

**Figure 1 ijerph-20-03521-f001:**
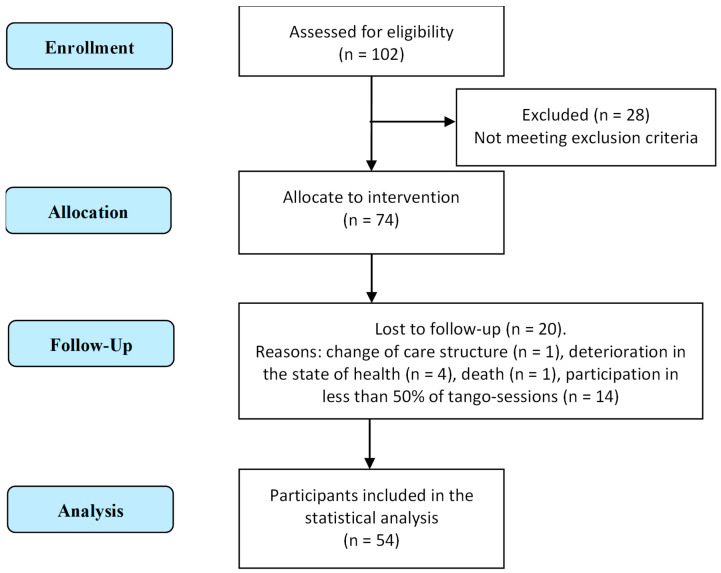
Flowchart for enrollment, allocation and follow-up of participants.

**Table 1 ijerph-20-03521-t001:** Overview of session content.

Session Components	Activities
Pre-warm-up	Organizing the seating arrangement in the room, greeting the participants, engaging in small talk.
Warm-up	Seated exercises to mobilize lower and upper limbs, head and trunk as well as singing to warm-up the voice and foster social connection.
Main part	Different aspects of tango therapy were practiced. This could include technical aspects, such as forward and backward walking, side-step, square, rectangle, as well as improvisation via spontaneous expression. The physical connection via ‘Abrazo’ (embrace) or other interactions were also an important aspect of the intervention.
Cool-down	Seated rituals, such as singing and breathing exercises.
Social exchange	The sessions were frequently followed by coffee and cake, or participants stayed to talk with each other and facilitators.

**Table 2 ijerph-20-03521-t002:** Songs frequently used.

Song Title	Artist
*Le plus beau tango du monde*	Tino Rossi
*Paloma*	Tino Rossi
*Mon Amant de Saint Jean*	Lucienne Delyle
*Voulez-Vous Danser Grand’mère*	Lina Margy
*Vous Permettez Monsieur*	Salvatore Adamo

**Table 3 ijerph-20-03521-t003:** Participant characteristics.

Variables	Total (n = 54)	SMCI (n = 41)	MNCI (n = 13)
Sex, females, n (%)	42 (78)	32 (78)	10 (77)
Age (years), M ± SD	84.9 ± 6.7	84.9 ± 6.8	84.9 ± 6.9
Charlson Index, M ± SD	5.6 ± 1.6	5.4 ± 1.2	6.4 ± 2.6
MMSE, score (range 0–30) M ± SD	14.5 ± 7.4	11.3 ± 5.1	24.5 ± 2.5
Katz Index, score (range 0–6) M ± SD	4.5 ± 1.3	4.3 ± 1.3	5.4± 0.8

SMCI: severe or moderate cognitive impairment; MNCI: mild or no cognitive impairment; M: mean; SD: standard deviation; MMSE: mini mental state examination.

**Table 4 ijerph-20-03521-t004:** Variables of physical abilities, functional capacities and quality of life before and after a 3-month tango program.

Variables		Pre-Test M ± SD	Post-Test M ± SD	*p*-Value
SPPB	Balance subscore (0–4)	2.1 ± 0.9	2.3 ± 1	0.221
	Gait speed subscore (0–4)	2 ± 1	2 ± 1.1	0.725
	Sit to stand subscore (0–4)	1.2 ± 1.2	1.2 ± 1.2	0.862
	Total score (0–12	5.3 ± 2.4	5.4 ± 2.6	0.876
Katz Index, score (range 0–6) M ± SD		4.5 ± 1.3	4.3 ± 1.2	0.253
QoL-AD, score (range 13–52)	Participant	33.8 ± 5.4	34.3 ± 5.7	0.172
	Caregiver	31.9 ± 4.5	33 ± 4.2	0.103
	Weighted composite score	33.1 ± 4.6	34 ± 4.5	0.030 *

M: mean; SD: standard deviation; SPPB: short physical performance battery; QoL-AD: quality of life in Alzheimer’s disease. * *p* < 0.05 based on two-sample t-test or Wilcoxon signed-rank test to test for differences before and after intervention.

**Table 5 ijerph-20-03521-t005:** Subgroup analyses according to the score in the mini-mental state examination.

Variables		SMCI (n = 41)		MNCI (n = 13)		Intergroup Comparison		Intragroup Comparison	
		T1	T2	T1	T2	T1/T1	T2/T2	SMCI	MNCI
SPPB	Balance (0–4)	2.05 ± 0.8	2.2 ± 0.9	2.4 ± 1.2	2.7 ± 1.2	0.426	0.075	0.422	0.305
	Gait speed (0–4)	1.9 ± 0.9	1.7 ± 0.9	2.5 ± 1.4	2.7 ± 1.3	0.171	0.020 *	0.486	0.527
	Sit to stand (0–4)	1 ± 0.9	1 ± 1	2 ± 1.6	1.9 ± 1.4	0.042 *	0.026 *	0.966	0.725
	Total (0–12)	4.9 ± 1.8	4.8 ± 1.9	6.8 ± 3.5	7.2 ± 3.6	0.09	0.025 *	0728	0.751
Katz Index		4.3 ± 1.3	4 ± 1.2	5.4 ± 0.8	5.3 ± 0.9	0.003 *	0.001 *	0.25	0.914
QoL-AD (13–52)	Participant	33.3 ± 5.1	34 ± 4.9	35 ± 6.1	35.2 ± 7.4	0.472	0.776	0.122	0.858
	Caregiver	31.4 ± 4.2	32.4 ± 4.2	33.3 ± 5	34.7 ± 3.9	0.238	0.203	0.205	0.441
	Composite score	32.4 ± 4.2	33.6 ± 4	34.4 ± 5.1	35 ± 5.7	0.368	0.434	0.038 *	0.552
Intervention Attendance		7.8 ± 1.7		8.7 ± 1.6		0.084			
EVIBE		4.5 ± 0.5		4.7 ± 0.4		0.087			

SMCI: severe or moderate cognitive impairment; MNCI: mild or no cognitive impairment; T1: before intervention; T2: after intervention; SPPB: short physical performance battery; QoL-AD: quality of life in Alzheimer’s disease. * *p* < 0.05 based on Wilcoxon signed-ranked test to test for differences before and after intervention and based on Mann–Whitney U test for differences between categories.

## Data Availability

The datasets generated during and/or analyzed during the current study are available from the corresponding author on reasonable request.

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
