# Peer review of "Tango-Therapy Intervention for Older Adults with Cognitive Impairment Living in Nursing Homes: Effects on Quality of Life, Physical Abilities and Gait"

_ijerph, 2023, doi:10.3390/ijerph20043521_

Round 1

Reviewer 1 Report

MATERIAL AND METHODS

(lines 92-93) What was the role and importance of the Dance Movement Therapist, in the context of the study? Why did the Therapist and the Musician only participate in half of the sessions?

(lines 106-111) The data collected by this instrument are not presented in the results section. Why? 

(lines 111-113) Considering this instrument is a scale and, therefore, has as an outcome an ordinal variable, would it not have been more appropriate to resort to an intra-class correlation coefficient, instead of Cohen's kappa?

(lines 145-146) Why was the Spearman's test used and not Pearson's? Where is the data distribution normality test described, which should precede the correlation test and dictated the option for Spearman?

RESULTS

Table 3. Charlson Index refers to which evaluated parameter? There is no reference to it in the methodology.

(lines 173-176) No statistically significant improvements were observed, in an intra-variable analysis, that is, in a longitudinal perspective, but it was hypothesized if there were cross-sectional (inter-variable) differences, that is, the subjects' performance in the SPPB tasks could, for example, be correlated (directly or inversely) or not with the ability to perform tasks of daily living (Katz Index) and/or with the perception of the subject's quality of life?

Figures 2 and 3. Due to lack of statistical significance, it makes no sense to present the graphical representation of the data.

Author Response

Thank you for taking the time to review and comment on our manuscript. We found the advice constructive and have incorporated many of the suggestions into our revision. We’ve responded to each comment individually below.

Thank you again for your thoughtful comments.

Sincerely,

LB

MATERIAL AND METHODS

  • (lines 92-93)What was the role and importance of the Dance Movement Therapist, in the context of the study? Why did the Therapist and the Musician only participate in half of the sessions?

Response: The presence of the DMT is very important because the goal of these tango interventions is not the entertainment of older people but to provide an alternative therapy to prevent functional decline and to improve QoL. We have expanded the intervention section in order to improve the explanation

The therapist and the musician only participate in half of the sessions for two reasons: the cost of their participation (funding was limited to one session) and -importantly- to engage consistently the care staff, who previously received training in Therapeutic Tango from Burgundy University to carry out the intervention by themselves.

  • (lines 106-111)The data collected by this instrument are not presented in the results section. Why? 

Response: The results of EVIBE are presented in line 231 and in Table 4. 

  • (lines 111-113)Considering this instrument is a scale and, therefore, has as an outcome an ordinal variable, would it not have been more appropriate to resort to an intra-class correlation coefficient, instead of Cohen's kappa?

Response: Thank you for the suggestion. We have provided the intra-class correlation coefficient, instead of Cohen's kappa (lines 159-160).

  • (lines 145-146)Why was the Spearman's test used and not Pearson's? Where is the data distribution normality test described, which should precede the correlation test and dictated the option for Spearman?

Response: We added a phrase about the Shapiro-Wilk test, used to determine data distribution normality (line 201). Spearman was used because most of the variables are non-parametric. However, we did not find correlations that were relevant to the main objective of the study, so we decided to remove the sentence about correlation in the statistical analysis section.

RESULTS

  • Table 3.Charlson Index refers to which evaluated parameter? There is no reference to it in the methodology.

Response: Thank you for this comment. We corrected this omission and described this variable in lines 148-149.

  • (lines 173-176)No statistically significant improvements were observed, in an intra-variable analysis, that is, in a longitudinal perspective, but it was hypothesized if there were cross-sectional (inter-variable) differences, that is, the subjects' performance in the SPPB tasks could, for example, be correlated (directly or inversely) or not with the ability to perform tasks of daily living (Katz Index) and/or with the perception of the subject's quality of life?

Response: Thank you for your comment. We think it is very interesting to look for the correlations between physical abilities, ADL and QoL in older people with CI. However, we found no correlations that were relevant to the subject of our article.

  • Figures 2 and 3.Due to lack of statistical significance, it makes no sense to present the graphical representation of the data.

Response: We agree with your comment. We have deleted the figures

Reviewer 2 Report

This manuscript entitled “Tango-therapy intervention for older adults with cognitive impairment living in nursing homes. Effects on quality of life, physical abilities and gait” primarily aimed to analyze the feasibility and the efficiency of tango-interventions in older adults living in nursing homes. To enhance the quality of the manuscript, revise suggestions are given below.

The author should show more examples of the Tango-therapy on other types of patients during the introduction.

I believe the test procedure and outcome that the author performed is not enough to show how this intervention works. Additionally, I don’t think these outcome parameters can reflect if this intervention is negative or positive. Please, can the author clarify this?

Please provide how you performed the gait test on the participants. It should be a procedure figure of the test.

Please provide the ethic approval number.

Author Response

Thank you for taking the time to review and comment on our manuscript. We found the advice constructive and have incorporated many of the suggestions into our revision. We’ve responded to each comment individually below.

Thank you again for your thoughtful comments.

Sincerely,

LB

  • The author should show more examples of the Tango-therapy on other types of patients during the introduction.

Response: Thank you for this comment. We have expanded the introduction providing more information about tango therapy in patients with Parkinson's disease, which is currently the most recognized application. The intervention approach is basically the same.

  • I believe the test procedure and outcome that the author performed is not enough to show how this intervention works. Additionally, I don’t think these outcome parameters can reflect if this intervention is negative or positive. Please, can the author clarify this?

Response: Your comment is very interesting. Some studies evaluate the psychosocial impact of dance on similar populations. However, our primary goal was to focus on the effect on physical abilities. Before performing our study, we performed a literature review (Bracco et al. 2021, http://cms.galenos.com.tr/Uploads/Article_49305/EurJGeriatricGerontol-3-134-En.pdf ). We based our protocol on the available scientific evidence. Nevertheless, as we explain in the paper, even if most studies used the SPPB to assess the physical performances in older adults with CI, this instrument may not be sensitive enough to measure outcome change, because minimal cognitive performance is required to understand instructions and respond appropriately. However, outcomes such as the capacity to perform activities of daily living is a very appropriate parameter, because this provides a piece of real information about the daily life of the participants. 

It was not our goal here to show how the intervention works (mechanism or change factor study), but that it works (outcome study). The how may follow in the next study.

  • Please provide how you performed the gait test on the participants. It should be a procedure figure of the test.

Response: 4-meters gait speed is part of the Short Physical Performance Battery.  We have improved the methods section and provided a succinct description and reference of this test. It is a widely used test to evaluate gait speed in older adults and it is an indicator of gait quality. The second indicator of gait quality was technical aids.

  • Please provide the ethic approval number.

Response: (CERUBFC 2021-09-15-026) (Line 107 and 428)

Reviewer 3 Report

This is an interesting study, however, this study has some limitations and gaps that need to be better explained and written by the authors, below are my considerations:

ABSTRACT

- I suggest that the authors add the p-value in the results section of the summary, of the outcomes that did not show significant differences.

INTRODUCTION

- I believe that the second paragraph can be improved, the paragraph has many interruptions that stop and start with “Dance could be” “Dance is” “Dance may” being repeated several times, this leaves the reading tiring for readers. I suggest improving the paragraph, giving the text a better fluidity.

- The authors mention that Tango can improve the outcomes they intend to analyze, and cite study 14, however, they do not mention what the present study presents as different from study 14, the same occurs with studies 23-25, I suggest adding , which again this study adds to the topic.

METHODS

- The authors mention the interventions, but do not explain where and how they decided to opt for these parameters in the interventions, I suggest adding where the authors took or decided to include these parameters.

- The same occurs in the steps of Tango, where and why the authors chose these parameters, they need to be justified in the study.

RESULTS

- The authors must include in Table 4 the test that was used in the comparison before and after the interventions.

DISCUSSION

- The objectives of the end of the introduction and the beginning of the discussion are different, I suggest standardizing the objectives.

- In the discussion the authors mention that more studies are needed on the topic, and this is very vague. Authors need to give more targeted information to authors researching this topic. What are the strengths observed by the authors in the present study, which should be continued? What are the difficulties faced by the authors that future studies should be more careful and attentive to? Could other outcomes be used? It is this type of more precise and targeted information that needs to be mentioned by the authors, and not just that more studies need to be carried out, this is scientifically poor information.

- An important gap noticed in this study was the discussion of how the Tango intervention can improve quality of life, or outcomes. Neurophysiologically, how did the Tango intervention reach this result? Authors need to explain the finding of their study.

- The authors mention the concern with falls in the introduction of the study, however, no outcome in this sense was measured by the authors. I suggest that authors suggest this for future trials in this manuscript and, if possible, include this outcome in their next trials.

Author Response

Thank you for taking the time to review and comment on our manuscript. We found the advice constructive and have incorporated many of the suggestions into our revision. We’ve responded to each comment individually below.

Thank you again for your thoughtful comments.

Sincerely,

LB

This is an interesting study, however, this study has some limitations and gaps that need to be better explained and written by the authors, below are my considerations:

ABSTRACT

  • I suggest that the authors add the p-value in the results section of the summary, of the outcomes that did not show significant differences.

 Response: Thank you for the suggestion which has been implemented in the summary of the results section.

INTRODUCTION

  • I believe that the second paragraph can be improved, the paragraph has many interruptions that stop and start with “Dance could be” “Dance is” “Dance may” being repeated several times, this leaves the reading tiring for readers. I suggest improving the paragraph, giving the text a better fluidity.

Response: Agree, the paragraph has been revised. 

  • The authors mention that Tango can improve the outcomes they intend to analyze, and cite study 14, however, they do not mention what the present study presents as different from study 14, the same occurs with studies 23-25, I suggest adding , which again this study adds to the topic.

Response: Thank you for the comment. We added an explanation about the sample study of reference 14 and we explain at the end of the introduction why our project is pertinent and original (lines 81-88)

METHODS

  • The authors mention the interventions, but do not explain where and how they decided to opt for these parameters in the interventions, I suggest adding where the authors took or decided to include these parameters (109-139)

Response: Thanks, we have expanded on the description and implementation of the intervention. 

  • The same occurs in the steps of Tango, where and why the authors chose these parameters, they need to be justified in the study.

Response: We have expanded on the description and implementation of the intervention. 

RESULTS

  • The authors must include in Table 4 the test that was used in the comparison before and after the interventions.

Response: Thank you for the suggestion. We added it in Table 4 and Table 5.

DISCUSSION

  • The objectives of the end of the introduction and the beginning of the discussion are different, I suggest standardizing the objectives.

Response: Thank you for de suggestion. We changed the beginning of the discussion to standardize the objectives. 

  • In the discussion the authors mention that more studies are needed on the topic, and this is very vague. Authors need to give more targeted information to authors researching this topic. What are the strengths observed by the authors in the present study, which should be continued? What are the difficulties faced by the authors that future studies should be more careful and attentive to? Could other outcomes be used? It is this type of more precise and targeted information that needs to be mentioned by the authors, and not just that more studies need to be carried out, this is scientifically poor information.

Response: Thank you for this comment. In lines 324-326 we have added an explanation about future investigations to understand the mechanism of the improvement of QoL through Tango. 

Also, in lines 397-402 we organized a paragraph about future research.

  • An important gap noticed in this study was the discussion of how the Tango intervention can improve quality of life, or outcomes. Neurophysiologically, how did the Tango intervention reach this result? Authors need to explain the finding of their study.

Response: Thank you for this comment. Even though the mechanisms involved in improving the quality of life through music and dance are still uncertain, we have added some lines in the discussion on the need for further investigations in this direction (lines 304-311).

  • The authors mention the concern with falls in the introduction of the study, however, no outcome in this sense was measured by the authors. I suggest that authors suggest this for future trials in this manuscript and, if possible, include this outcome in their next trials.

Response: Thank you, we added this suggestion at the end of the discussion. 

Reviewer 4 Report

The authors analyzed effect of tango therapy treatments on older adults. Experimental settings need improvement and results and methods lack novelty. Here are my comments:

·         There are other studies that analyzed Tango interventions, what’s the difference in this study? The authors should add this discussion in introduction section.

·         Novelty and objectives of this study should be explained in a more detailed way in introduction section.

·         Information about participants can be increased. Maybe a table can be added that summarizes participants demographic data.

·         How can authors be sure that this intervention has positive impact since there aren’t any control groups? The other studies included control groups in their experiments.

·         Obtained results show very little improvement. But authors reported that this therapy is very successful and feasible, these statements create a contradiction.

Author Response

Thank you for taking the time to review and comment on our manuscript. We found the advice constructive and have incorporated many of the suggestions into our revision. We’ve responded to each comment individually below.

Thank you again for your thoughtful comments.

Sincerely,

LB

The authors analyzed effect of tango therapy treatments on older adults. Experimental settings need improvement and results and methods lack novelty. Here are my comments:

  • There are other studies that analyzed Tango interventions, what’s the difference in this study? The authors should add this discussion in introduction section.

Response: Thank you for this comment. This is the first study analyzing the feasibility and effectiveness of tango in older people with CI living in nursing homes. As you suggest, we added an explanation about it in the introduction. 

  • Novelty and objectives of this study should be explained in a more detailed way in introduction section.

Response: we have added explanations about novelty and objectives in the introduction (lines 81-88).

  • Information about participants can be increased. Maybe a table can be added that summarizes participants demographic data.

Response: Participants demographic data are presented in Table 3 (age, sex, cognitive performance, comorbidities, abilities to carry out ADL).

  •  
  • How can authors be sure that this intervention has positive impact since there aren’t any control groups? The other studies included control groups in their experiments.

Response: we agree with your comment since the lack of control groups is one of the limits of our study. However, we consider that this study is one important piece of the building that provides evidence on non-pharmacological interventions, since there is hardly any evidence on their efficacy in this population.

  • Obtained results show very little improvement. But authors reported that this therapy is very successful and feasible, these statements create a contradiction.

Response: Thank you for this comment. We would like to argue that stating a success is not an overstatement here: Generally, the tendency of institutionalized advanced-age persons, with a high degree of comorbidities, dependency and cognitive impairment, is towards functional deterioration. For this reason, we consider that a stabilization of the physical capacities and an improvement of the quality of life is a great success. But, for sure, future research with a control group is necessary to clarify the effect of this non-pharmacological intervention. We reformulated the discussion and the conclusions in this way.

Reviewer 5 Report

Dear Authors, 

Thanks for your efforts in testing the feasibility of the intervention of interest. And thanks again for addressing the weaknesses of the design, however, this would be the initial testing of the intervention, and I hope you test it next using the RCT design. 

Please add a piece of information regarding the validity and reliability of QoL measure in previous studies, and it would be better if you also include the reliability of the QOL in the current sample. 

Author Response

Thanks for your efforts in testing the feasibility of the intervention of interest. And thanks again for addressing the weaknesses of the design, however, this would be the initial testing of the intervention, and I hope you test it next using the RCT design. 

Please add a piece of information regarding the validity and reliability of QoL measure in previous studies, and it would be better if you also include the reliability of the QOL in the current sample. 

Response:

Thank you for taking the time to review and comment on our manuscript. We found the advice constructive and have incorporated your suggestion into our revision.

We work at the moment in a RCT in order to deepen the issues that emerge from this first study without a control group.

We added validity and reliability information about QoL-AD (lines 190-197).

Thank you again for encouraging us to continue our work.

Sincerely,

LB

Round 2

Reviewer 2 Report

this manuscript can be accepted

Reviewer 3 Report

I congratulate the authors for the excellent work and for the changes in the manuscript, in this way, it became clearer for readers to interpret the text and how the study was developed.

Reviewer 4 Report

Thank you for submitting your revised manuscript. The manuscript has been significantly improved since the first submission and I'm overall satisfied with the corrections.